# The Bio-Patina on a Hypogeum Wall of the Matera-Sassi Rupestrian Church “San Pietro Barisano” before and after Treatment with Glycoalkaloids

**DOI:** 10.3390/molecules28010330

**Published:** 2022-12-31

**Authors:** Francesco Cardellicchio, Sabino Aurelio Bufo, Stefania Mirela Mang, Ippolito Camele, Anna Maria Salvi, Laura Scrano

**Affiliations:** 1Institute of Methodologies for Environmental Analysis, Italian Research Council (CNR), Contrada Loya, 85050 Tito, Italy; 2Department of Sciences, University of Basilicata, Via dell’Ateneo Lucano 10, 85100 Potenza, Italy; 3Department of Geography, Environmental Management and Energy Studies, University of Johannesburg, Johannesburg 2092, South Africa; 4School of Agriculture, Forestry and Environment, University of Basilicata, 85100 Potenza, Italy; 5Department of European Cultures, University of Basilicata, 75100 Matera, Italy

**Keywords:** glycoalkaloids, bio-cleaning, XPS, SEM/EDS, cultural heritage, San Pietro Barisano church, Matera-Sassi

## Abstract

The investigation focused on the deterioration of the walls in the hypogeum of “San Pietro Barisano” rupestrian church, located in the Matera-Sassi (Southern Italy), one of the UNESCO World Heritage sites. The study evaluated the biocide activity of a mixture of natural glycoalkaloids (GAs) extracted from the unripe fruit of *Solanum nigrum* and applied to clean a hypogeum wall surface in the church affected by bio-patinas. The analyzed bio-patina, collected before treatment and, at pre-established times, after treatment, showed changes in chemical composition detected by XPS, accompanied by visible discoloration and biological activity variation. The biocidal action of the glycoalkaloids mixture, directly employed on the wall surface, was effective after about four weeks for most bio-patina colonizers but not for the fungal species that can migrate and survive in the porosities of the calcarenite. Consequently, the cleaning procedure requires the integration of fungicidal actions, combined with the consolidation of the surfaces, to obtain complete bioremediation and avoid subsequent biological recolonization. SEM images and associated microanalysis of pretreated bio-patina have revealed the biocalcogenity of some autochthonous microorganisms, thus preluding to their eventual isolation and reintroduction on the wall surface to act as consolidants once the bio-cleaning phase has been completed.

## 1. Introduction

In recent decades, historical and artistic heritage has undergone greater degradation than in the past due to the synergistic action of atmospheric pollution, climate change, and increased biological contamination. The conservation of this heritage also requires the development of innovative, effective, and, at the same time, non-expensive protection strategies. In particular, the biodeterioration of stone materials is related to the biological activity of colonizing microorganisms favored by environmental factors such as temperature, humidity, and lighting. The degradation process begins with surface alterations that produce unsightly and chromatic stains. After the first phase of settlement of primary microorganisms, biodegradation can considerably influence the integrity of the materials with negative consequences on their conservation [1,2,3,4]. Photoautotrophic microorganisms are the primary colonizers that can find suitable conditions to secrete exopolysaccharides, alginates, and other compounds useful for binding to the rock. The constitution of this biofilm then favors the settlement of secondary colonizations and heterotrophic microorganisms [5,6,7,8]. In recent years, studying biodegradation phenomena in sites of cultural interest has also been one of the activities carried out within the project “Smart Cities, Communities and Social Innovation, SCN_00520”, funded by the Italian Ministry of Research and University. The Sassi of Matera (Matera, Basilicata Region, Southern Italy) and the rupestrian churches carved into the limestone have represented important study sites for the extensive degradation phenomena of the surfaces also due to biological colonization [9,10]. In particular, attention was paid to the church of San Pietro Barisano (Figure 1), an example of a typical architectural structure of the Matera-Sassi system. The church also has a hypogeum on the lower level without natural light and enough air exchange. The climatic conditions of the hypogeum, dedicated in the past to the putrefaction of corpses before the final burial in the ossuary (*Putridarium*), were similar but extreme to those of the upper church with a relative humidity of the surfaces above 90% and an ambient temperature between 11 and 17 °C [9]. In this environment, the microclimatic conditions provide an ideal “niche” for developing photosynthetic organisms capable of exploiting the spectral emission of artificial lighting. Consequently, suitable conditions are created for the development of photoautotrophic organisms such as cyanobacteria, green filamentous algae, diatoms, and mosses responsible for forming biofilms on surfaces. At the same time, the products of photosynthesis provide nourishment for the settlement of heterotrophic species such as fungi and bacteria, the development of which produces an increase in biological activity on the colonized surface [10,11]. Biodegradation due to the development of lichens, bacteria, and fungi was also found in places of the upper church due to the high relative humidity and water infiltrations produced by the high porosity of the calcarenite. In Figure 2, for example, particular black crusts can be seen on the altar of St. Joseph [12,13]. For the elimination of biological patinas on the surfaces of historical-artistic artifacts, up to now, both inorganic compounds (such as sodium hypochlorite, hydrogen peroxide, and active chlorine) and organic compounds (such as formaldehyde esters, methyl-phosphates, chloramines, etc.) have been used [14]. These substances can be harmful to both humans and the environment and can also cause damage to architectural materials [15]. The present research work, therefore, was aimed at finding innovative and ecological approaches based on the use of biocides derived from natural compounds, which can interfere, at the molecular level, with the microbial communication system called “quorum sensing” and inhibit the initial phase of microbial biofilm formation [16,17]. Among the natural biocides, glycoalkaloids deserve attention. They are secondary metabolites produced by plants of the *Solanaceae* family, which use them as chemical defenses against pathogens such as fungi, bacteria, and viruses. A series of investigations on glycoalkaloids isolated from different species of plants of the genus *Solanum* has demonstrated antifungal activity [18,19,20] and antimicrobial action [21] of these compounds. Some glycoalkaloids have revealed antiviral activity. Glycoalkaloids such as solamargine and solasonine can also inactivate various forms of herpes [22]. The solasodine extracted from the flowers of *Solanum dulcamara* also inhibits the growth of *Escherichia coli* and *Staphylococcus aureus*. In this work, the glycoalkaloids’ crude extract (containing approximately 510 μM solamargine and 460 μM solasonine as main components) obtained from the berries of *Solanum nigrum* was tested in the hypogeum of San Pietro Barisano for bio-cleaning purposes. The experiment was carried out on a portion of the wall with an extensive biological patina. Bacterial and fungi species on the internal walls were identified with the final aim of choosing the best intervention strategy [13,17,23,24,25]. The disappearance of the biological patina was evaluated not only by a visual investigation but also by surface analytical techniques such as XPS photoelectron spectroscopy [26,27], scanning electron microscopy (SEM), and energy dispersive X-ray spectroscopy (EDS).

## 2. Results and Discussion

### 2.1. XPS Analysis

On the horizontal wall of the hypogeum, chosen as the study area, a sample of surface patina was taken and subjected to XPS analysis before treatment with the glycoalkaloid mixture. The analyses of this sample were compared with those of the samples taken after treatment. Figure 3 shows the XPS wide spectra and the peaks of interest related to the samples taken pre- and post-application of the glycoalkaloid mixture for four weeks.

The most representative elements in the labeled peaks were acquired at high resolution to proceed with the semi-quantitative analysis via curve-fitting and to derive the chemical composition of the patina before and after treatment with glycoalkaloids. The C1s region is the first to be analyzed (Figure 4) as it contains the carbonate peak, set at 290.0 eV from previous analysis of calcarenite samples and used as an internal reference to correct the shift on binding energies (BEs, eV) due to surface charging [28]. In addition, it is the region most contributive in discerning the organic components associated with the bio-patina’s biological activity and their intensity variation after the treatment with glycoalkaloids (Table 1 and Table 2).

Comparing the C1s carbon regions before and after the treatment with the biocide allows us to detect the decrease of the organic components related to the biological activity [29]. Indeed, it is possible to observe the disappearance of the peak due to calcium oxalate and the increase of the signal associated with the carbonate. These changes confirm the gradual removal of the biological patina, which initially covered the wall and highlighted the calcarenite component, following the biocidal action of the glycoalkaloids spread on the surface and left to act for four weeks. Figure 5 illustrates the Ca2p region relating to the pre- and post-treatment, and Table 3 reports the assignments of the various peaks. Noteworthy, the Ca2p region remains unchanged in shape, within the limits of energy resolution with achromatic sources.

Comparing the peaks of pre- and post-treatment and the related normalized areas, calcium level increases due to the rise in the percentage of carbonate. The growth of the calcium carbonate signal demonstrates the cleaning effectiveness; however, the XPS analysis results show the incomplete removal of the surface patina’s organic components, which may also include calcium alginate often found in biofilms of the colonizing organisms [30]. Alginate is a component of the cell walls of algae. In this work, the presence of microalgae (diatoms) was confirmed on the hypogeum walls by the SEM/EDS investigation illustrated below.

The O1s regions centered at 533.0 and 532.2 eV, respectively, before and after treatment, include a set of contributions due to the detected oxygenated compounds (C-O, O-H, SiOx bonds, phosphates, sulfates, metal oxides, etc.), some of which are so closely spaced in energy that they cannot be resolved by curve-fitting. However, the total area of O1s made it possible to evaluate the goodness of the assignments through the stoichiometric mass balance.

Table 4 shows the assignments of the peaks relating to minority components identified in the other regions analyzed. The regions refer to elements attributable to silicate and bioclastic compounds of the calcarenite and the metabolites present in the bio-patina, together with the carbonaceous components that make up the biofilm. Finally, after partial and overall mass balance cross-checking, the results were summarized in the pie charts in Figure 6, where the percentage distributions of the various chemical groups determined in the samples before and after treatment with the glycoalkaloid mixture are shown. There was a decrease in the organic carbon bound to the biological fraction present on the wall and a simultaneous increase in the carbonate fraction (from 21.6% to 33.8%) and silicate components typical of calcarenite rock (assuming as reference the chemical composition of the highly porous Gravina’s calcarenite, mainly composed of calcium carbonate, as calcite, phyllosilicates, and small percentages of quartz) [31]. The monitored changes confirm the gradual removal of some components linked to the biological activity of the patina covering the wall due to the biocidal action of the glycoalkaloids spread on the surface and left to act for four weeks.

Samples were taken from the wall in the treated area, even 1 and 2 weeks after application, to evaluate the biocide action according to the time of application. For brevity, below are only reported results for the C1s region, highlighting the most significant functional groups emphasized by the curve-fitting procedure. The comparisons of the C1s regions (Figure 7) are exciting and show that after a week of application, the peak 5 relative to the oxalate was no longer detectable. Moreover, the carbonate signal (peak 6) increased after the second week while the carbonaceous components were reduced.

It is essential to underline the disappearance of the peak centered at about 289 eV, attributable to oxalate, partially superimposed on the reference peak of carbonate at 290.0 eV but easily resolved by curve-fitting after the treatment.

However, the normalized areas Ca^2+^/CO_3_^2−^ ratio was never unitary, as theoretically expected for the stoichiometric calcium carbonate. The discordance from unity was decreased with the treatment time, and after four weeks, the Ca^2+^/CO_3_^2−^ ratio was nearly 0.90, reaching the lower limit of the uncertainty associated with XPS analysis, i.e., ±10% [28]. Thus, while the increase of the carbonate signal demonstrates the effectiveness of the cleaning action, the non-correct carbonate stoichiometry implies the bonding of calcium ions with other components of the surface patina, which can also include calcium alginate produced in the biofilm by the colonizing microorganisms, anchored on carbonate stones [5,6]. 

The carboxylic (COO^−^)_2_ interchain crosslinking with Ca^2+^ and the participation of alcohol and ether groups of the alginate chain in the coordination sphere are reported in the literature [32] together with the correspondent XPS assignments. These correspondences (Ca2p_3/2_ 347.7 and C1s 290.5 eV, practically superimposed to CaCO_3_ positions in both regions) were experimentally confirmed with the XPS acquisition of an alginate-based hydrogel, produced in our laboratory after a prolonged interaction with a Carrara marble surface [33].

Figure 8 shows the color variation of the surficial patina during the four weeks of treatment. During this period, the bio-patina reduced and simultaneously underwent chromatic variations until the initial green color almost wholly disappeared. This confirms that the producing-patina microorganisms were eliminated, demonstrating the effectiveness of the treatment after four weeks, as also shown by the experimental data obtained with the XPS analysis.

### 2.2. Biological Analysis

From the bio-patina of the San Pietro Barisano rupestrian church hypogeum, several bacterial and fungal species were isolated and further identified by their morphological key features and molecularly recognized by PCR amplification and sequencing of the two Internal Transcribed Spacers and 5.8 S gene (ITS1-5.8S-ITS2) from the nuclear ribosomal RNA (for fungi) and 16S gene (for bacteria), as shown in Table 5 and Figure 9. All fungal and bacterial nucleotide sequences were deposited in the GenBank (NCBI) under the following accession numbers: *Botryotrichum atrogriseum* (OP888461-OP888463)*; Penicillium chrysogenum* (OP890579-OP890581); *Talaromyces pinophilus* (OP894917-OP894919) and *Cladosporium herbarum* (OP890587-OP890590). The sequences of bacteria also received their accession numbers: *Staphylococcus warneri* (CP003668.1); *Brevibacillus* (SRR17297603); *Bacillus cereus* (CP020803.1); *Bacillus mycoides* (GCA_000832605); *Bacillus firmus* (SAMEA4076706).

The complex biological colony is typical of humid and poorly lit environments such as hypogea, caves, and catacombs [11,34,35]. In these environments, the organisms find the conditions suitable for their growth: humidity, organic material produced by primary or decomposing organisms, artificial lighting, etc. Among the bacterial species identified, some are toxic species, such as *B. cereus*, a pathogenic bacterium that produces toxins responsible for food poisoning, and *B. mycoides,* capable of causing diseases in some organisms. Among the fungal species, *P. chrysogenum*, a fungus belonging to the *Aspergillaceae* family, is known to be a pathogen that forms blue or grey-green mold. Many fungi identified, such as *T. pinophilus*, have high enzymatic activities capable of degrading various substrates. Species of the genus *Talaromyces* produce enzymes beneficial for degrading biomass and secondary metabolites. However, these enzymes are still poorly characterized due to the lack of complete genetic information. To verify the biocidal action on the fungi found on the hypogeal walls, the glycoalkaloids extracted from *S. nigrum* were tested on *B. atrogriseum* and *T. pinophilus* at the following concentrations: 100, 75, 50, and 25%. The in vitro results showed low biocidal activity for these fungal species due to a restored activity of the fungal colony four weeks after the treatment with glycoalkaloids [12]. Despite the antibacterial activity of glycoalkaloids demonstrated by previous findings [23,36], the studied extract is similarly in vivo characterized by low effectiveness towards fungal species. Under these conditions, the treatment could even increase fungal colonies that can use dead bacterial cells as a carbon source.

Fungal survival could also be favored by a possible migration inside the stone’s pores, given the extensive alveolarization, making biocidal action difficult.

The role of various microorganisms in biofilms is still a complex matter; e.g., it is notable the joint function of compounds such as calcium oxalate, which, although produced by the symbiosis of fungal species and oxalotrophic bacteria, can only be metabolized by the bacteria with the formation of calcium carbonate [37].

It is also worth emphasizing the lower efficacy of glycoalkaloids supported on gels [12]. However, gels, or sometimes cellulose pulp, have proved very useful within the Smart Cities project to work more precisely and avoid the deposition of other contaminants on the surface [31,38,39,40,41]. In the case of the hypogeum of San Pietro Barisano church, the experimentation with sodium alginate-based gel did not give satisfactory results due to degradation and liquefaction phenomena of the gel caused by the particular microclimatic conditions [12].

### 2.3. SEM/EDS Results

Electron microscopy investigations were conducted only on fragments already detached from the hypogeum walls under examination, thus examined before their treatment with glycoalkaloids. The SEM image of Figure 10 highlights the presence of microalgae on the hypogeum walls, in the specific case of diatoms, identified by the presence in the relative EDS spectrum of silicon which is a component of the shell of these organisms. The calcium carbonate crystals on the surface of the hypogeum walls do not have a regular rhombic shape and, as shown in Figure 11 along the same fragment, their arrangement is either linked to the chemical-physical equilibrium that is constantly established with the high humidity environments (central image and EDS) and to the activity of the microorganisms present in the patina as evidenced by the calcium carbonate deposits on the fungal hyphae (left image), produced by the fungi themselves.

## 3. Materials and Methods

### 3.1. Sampling

Due to its microclimatic conditions and its walls with extensive biological colonization, the hypogeum of San Pietro Barisano proved to be an ideal site for testing the biocidal efficacy of glycoalkaloids.

Figure 12 illustrates the portion of the wall chosen for the various tests.

A non–invasive sampling was performed according to the standard procedures foreseen by the Normal Recommendations—1/88 [42]. The samples were obtained by gently scraping the hypogeum walls before and after biocide treatment. With the aid of a spatula, samples were collected by removing the bio-patina without affecting the underlying calcareous substrate. The samples were homogenized and stored for XPS and microbiological analyses to determine the chemical composition of the bio-patina and the biological composition.

### 3.2. Glycoalkaloids Extraction

The glycoalkaloids used in this study were those extracted from unripe berries of *Solanum nigrum*, a spontaneous plant belonging to the *Solanaceae* family, using the method proposed by Cataldi et al. [43]. The freeze-dried berries were placed in an aqueous 1% acetic acid solution. To facilitate the contact between the plant tissue and the extraction solvent, the suspension was stirred for about 2 h and then centrifuged at 3000× *g* for 20 min at 4 °C. The pellet was suspended in 1% acetic acid, mixed, centrifuged, and filtered through a 0.22 µm nylon filter (Whatman, Maidstone). The extracts, characterized by liquid chromatography-mass spectrometry (LC-ESI-MS), contained similar amounts of the two main glycoalkaloids, solamargine and solasonine, and other less abundant components [17,23,36].

### 3.3. Surface Treatment of the Hypogeum Wall

A total of 500 mg of the extracted, lyophilized, and pulverized glycoalkaloids were added to 10 mL of water [17]. The solution was spread directly on the surface of the hypogeum wall, selected for the cleaning treatment, using a special soft brush. Since the hypogeum is an environment with high humidity, it was not necessary to cover the portion of the wall treated to prevent the applied solution from drying out. The surface treatment time was set at four weeks. To monitor the biocidal action of the glycoalkaloids over time, samples were also taken after 1 and 2 weeks. The changes in the bio-patina were followed based on the visual appearance and the results of the XPS analysis.

### 3.4. X-ray Photoelectron Spectroscopy (XPS)

The samples for XPS analysis were homogenized in an agate mortar and then pressed onto a double-sided copper tape, fixed adequately on a steel sample holder, to be safely introduced in the analysis chamber of the spectrometer. XPS spectra were acquired with a SPECS Phoibos 100-MCD5 spectrometer operating at 10 kV and 10 mA, in medium area (diameter = 2 mm) mode, using a MgKα (1253.6 eV) and AlKα (1486.6 eV) radiations. The pressure in the analysis chamber was less than 10^−9^ mbar during acquisition. Wide spectra were acquired in FAT (Fixed Analyzer Transmission) or FRR (Fixed Retarding Ratio) modes. High-resolution spectra were acquired in the FAT mode, with a constant pass energy of 9 eV and channel widths of 0.1 eV. They were “curve-fitted” using the Googly program, which allows the evaluation of intrinsic and extrinsic features of XPS spectra [44,45]. Peak areas and positions (Binding Energies, BE) as derived by the “curve-fitting” procedure were, respectively, normalized using proper sensitivity factors and referenced to the C1s carbonate carbon set at 290.0 eV [28]. The assignments of the corresponding chemical groups and the relative percentage compositions were derived from the analysis of standard compounds acquired in the laboratory and from the NIST X-ray Photoelectron Spectroscopy Database [46].

### 3.5. Biological Analysis

To characterize the biological colonization, investigations were conducted at the School of Agricultural, Forestry, Food and Environmental Sciences (SAFE) and Department of European and Mediterranean Cultures (DiCEM) laboratories of the University of Basilicata, Italy.

Samples of powdered bio-patinas, stored in sterile Eppendorf tubes, were suspended in a culture medium to isolate the bacterial strains and proceeded to the subsequent extraction and characterization of the DNA. For the description of bacterial species, the analyses were carried out according to the procedures indicated by Scrano et al. [17]. Bacterial colonies were isolated and characterized based on their different morphologies. The isolated strains were purified, and their DNA was used for molecular identification. Synthetic oligonucleotide primers fD1 (AGAGTTTGATCCTGGCTCAG) and rD1 (AAGGAGGTGATCCAGCC) were used to amplify the 16S rDNA. As previously reported, PCR mixture and amplification conditions were performed [23,47]. The PCR products were sequenced using Genetic Analyzer 3130xl (Applied Biosystems), and the GenBank and EMBL database performed the DNA similarity using the BLAST suite.

For fungal isolation, samples were collected in sterile vials containing 1 mL of bi-distilled water, stored in the refrigerator at 4 ◦C, and analyzed within 24 h. A 100 µL of fungal suspension was directly placed on Petri dishes containing “Potato Dextrose Agar” (PDA) added with kanamycin (1 mg/L) and streptomycin (1 mg/L). The plates were incubated in an incubator at 24 °C ± 1 °C in the dark for seven days. The pure fungal cultures (PFC) were used for further morphological and molecular investigations and preliminarily identified using a microscope (Axioscope, Zeiss-Germany) based on their macroscopic and microscopic characteristics. The identification of the fungal species was also carried out according to the technique described by Mang et al. [13].

Briefly, PFC mycelium was scraped from the colony’s surface and finely homogenized using liquid nitrogen to isolate the genomic DNA (gDNA). The last was extracted from approximately 100 mg of each sample with the NucleoSpin Plant II ™ kit (Macherey-Nagel, Germany) following the manufacturer’s instructions. The gDNA of each PFC was then subjected to Polymerase Chain Reaction (PCR) using ITS5 and ITS4 or Bt2a and Bt2b primers which amplified a fragment of the Internal Transcribed Spacer (ITS) and the beta-tubulin gene (TUB-2), respectively. PCR products were detected by 1.2% (w/v) agarose gel electrophoresis, pre-stained with SYBR Safe DNA Gel Stain (Invitrogen Inc., Carlsbad, CA, USA), and photographed with a camera under the transilluminator model Euro-Clone (Pero, Milan, Italy). The amplicons were sequenced by Sanger Dideoxy technology using the same primers as for PCR. All fungal nucleotide sequences obtained in this study were deposited in the Genbank (NCBI). They were also compared with those existing there using the Basic Local Alignment Search Tool (BLASTn) program to identify the fungal isolates at genus and species level [47]. For correct identification of the identified taxa, uncultivated fungal species and those of dubious labels found in the NCBI GenBank were excluded.

### 3.6. SEM/EDS Analysis

Scanning electron microscopy analyses were carried out on fragments detached from the hypogeum walls at the CNR—IMAA (Institute of Methodologies for Environmental Analysis of the National Research Council, Tito Scalo, Potenza) using a Scanning Electronic Microscope with Field Emission Source (FESEM), Zeiss Supra 40 model, equipped with Oxford INCA. Energy 350 Energy Dispersion Microanalysis System (EDS) and X-act S.D.D. (Silicon Drift Detector) LN2-free detector. This tool allows for obtaining high-resolution (nanometric) images using an “in-lens” detector of secondary electrons. Since these are non-conductive stone samples, a metallization operation with graphite was carried out to obtain a high-quality image. Each sample analyzed different areas at different magnifications, using an acceleration voltage of 15 kV and an aperture of 60 μm. Using SEM, images were obtained both from the secondary electrons, which give information on the surface morphology and from the retro-scattered electrons through which it is possible to recognize differences in the sample’s elemental composition. With the microanalysis system, it was also possible to identify these elements from their X-rays emission detected by EDS (energy dispersion spectrometry).

## 4. Conclusions

The biocidal activity of glycoalkaloids extract (mainly solasonine and solamargine), already demonstrated against some bacterial species, was verified by treating a degraded surface of the hypogeum of the San Pietro Barisano church. The glycoalkaloid solution was left in direct contact with the bio-patina for 4 weeks. 

The cleaning effectiveness was visually checked with the color change of the surface. Biological and electron microscopy analyses supplied information on the living components and the characteristics of the surfaces, which are made up of porous calcarenite and offer settlement sites for the various organisms. XPS photoelectron spectroscopy has proved to be an interesting analytical technique for evaluating organic and inorganic components present on the surface and studying the changes induced by the biocide treatment operated by using glycoalkaloids. The visual results show that the biocide acts on the bio-patina by reducing its color and thickness; from the XPS analysis, it was possible to demonstrate the reduction of organic compounds attributable to biological activity.

The results highlighted the removal of the bacterial load but not the effect on the fungal colonies.

Under these conditions, the treatment could increase fungal colonies at the expense of bacteria and use dead bacterial cells as a carbon source. Fungal survival is also favored by a possible migration inside the stone’s pores, given the extensive alveolarization present, which makes the biocidal action of the gel difficult.

One of the immediate research perspectives is optimizing the cleaning process by combining the antibacterial actions of glycoalkaloids with those of other natural extracts with antifungal properties possibly supported on a gel, also of natural origin. In this case, the particular microclimatic and humidity conditions present in the hypogeum of the church of San Pietro Barisano are factors to be considered.

The biocidal actions could be integrated with the bio-consolidation of the treated surface to confer resistance to the underlying support against new attacks by biodeteriogens. As reported in the literature and highlighted here by the electron microscopy analyses, some microorganisms can produce compounds chemically and structurally compatible with the carbonate stones by decreasing their porosity. The development of innovative bioremediation techniques based on native colonizing microorganisms capable of inducing carbonate precipitation and structural consolidation represented one of the macro-objectives of the SCN_0520 project.

## Figures and Tables

**Figure 1 molecules-28-00330-f001:**
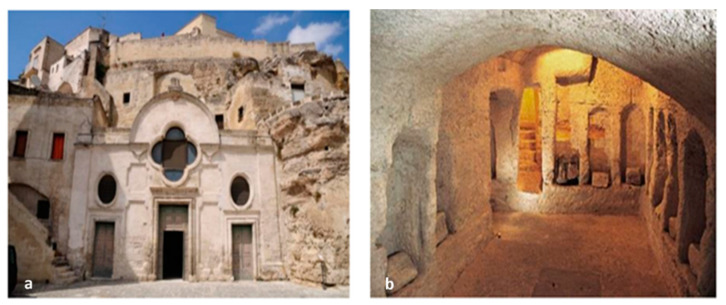
External facade (**a**) and hypogeum (**b**) of the San Pietro Barisano church.

**Figure 2 molecules-28-00330-f002:**
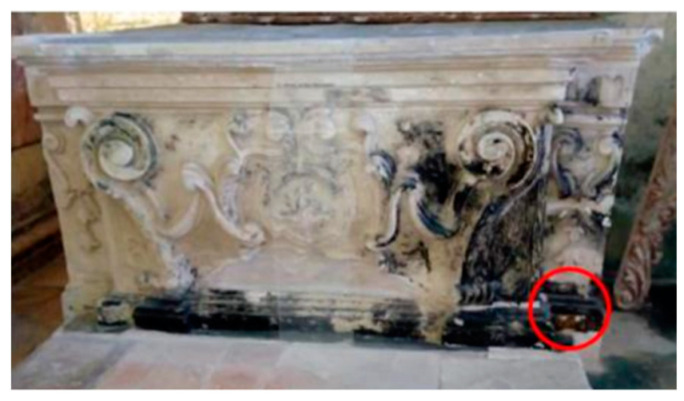
Black crusts on the altar of St. Joseph in the San Pietro Barisano church.

**Figure 3 molecules-28-00330-f003:**
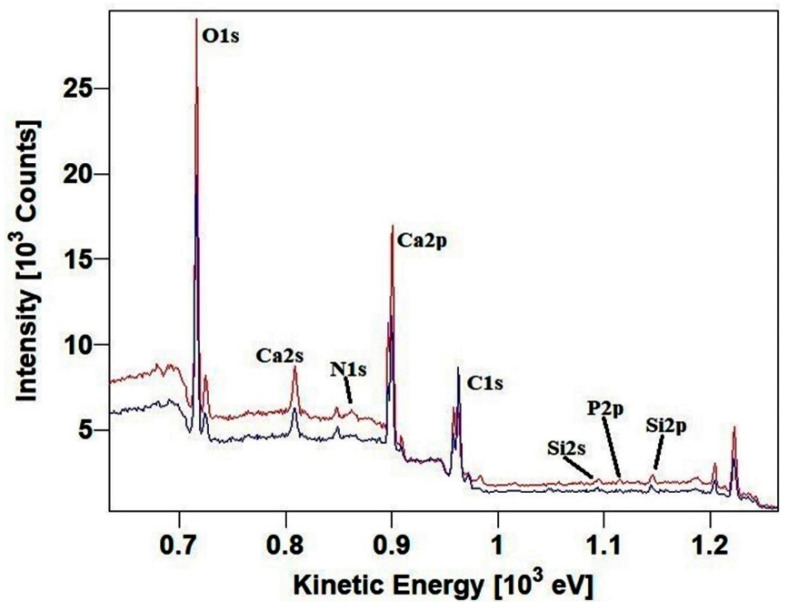
Wide spectra of the samples of the horizontal wall of the hypogeum: pre-(blue) and post-(red) application of glycoalkaloids extract.

**Figure 4 molecules-28-00330-f004:**
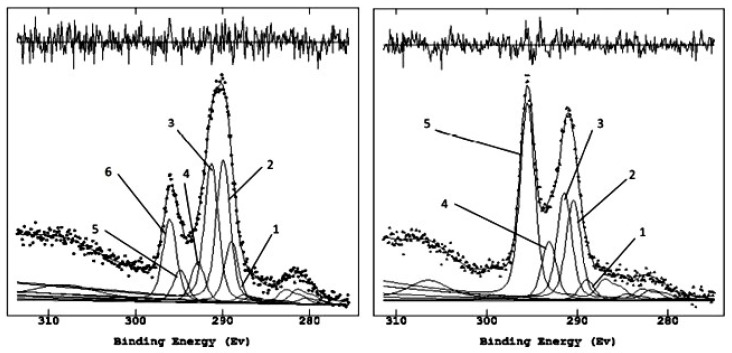
Curve fitting of the C1s region pre-(**left**) and post-(**right**)application of glycoalkaloids.

**Figure 5 molecules-28-00330-f005:**
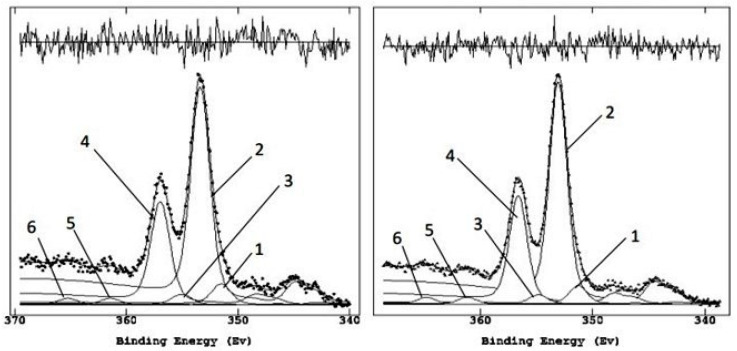
Curve fitting of Ca2p regions pre (**left**) and post (**right**) glycoalkaloids application.

**Figure 6 molecules-28-00330-f006:**
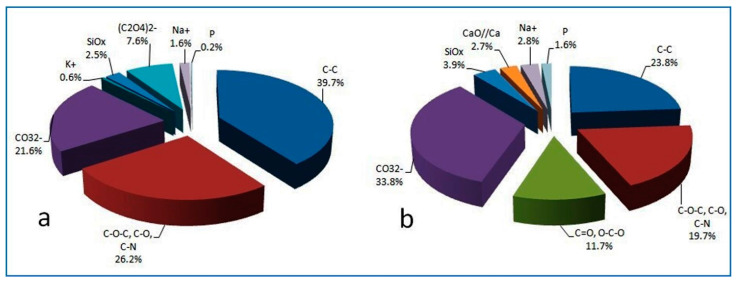
Percentage distribution of the chemical groups determined in the patina samples before (**a**) and after (**b**) treatment with glycoalkaloids.

**Figure 7 molecules-28-00330-f007:**
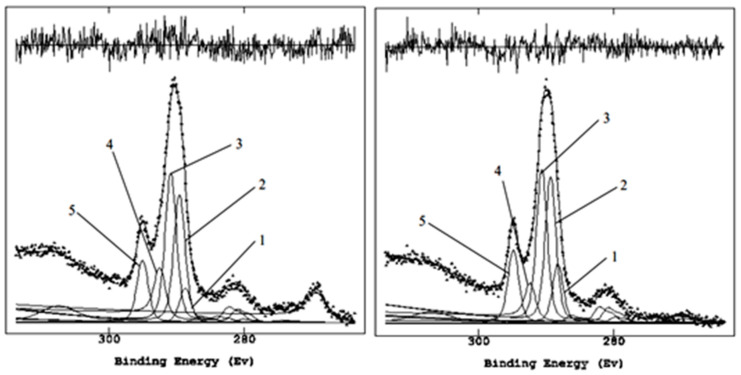
Curve fitting of C1s regions post-1 week (**left**), post-2 weeks (**right**).

**Figure 8 molecules-28-00330-f008:**
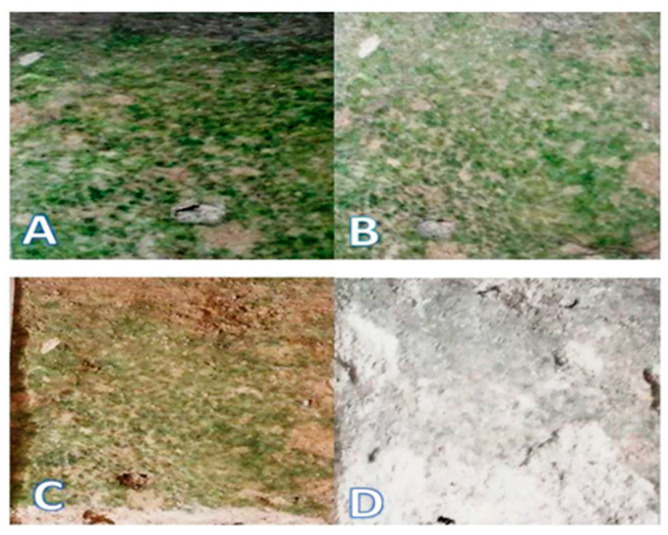
Color appearance of the bio-patina in the hypogeum of San Pietro Barisano during the treatment period with glycoalkaloids: (**A**) before the treatment; (**B**) post-1 week; (**C**) post-2 weeks; (**D**) post-4 weeks.

**Figure 9 molecules-28-00330-f009:**
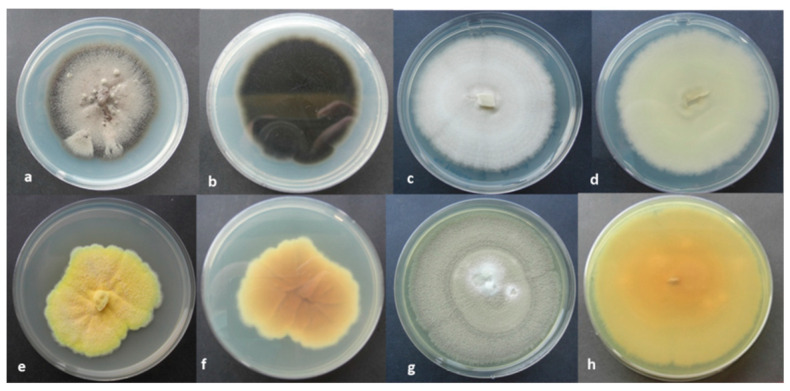
Fungal species isolated from hypogeum walls of the San Pietro Barisano rupestrian church on PDA media (**a**,**b** = *C. herbarium*; **c**,**d** = *B. atrogriseum*; **e**,**f** = *T. pinophilus*; **g**,**h** = *P. chrysogenum*).

**Figure 10 molecules-28-00330-f010:**
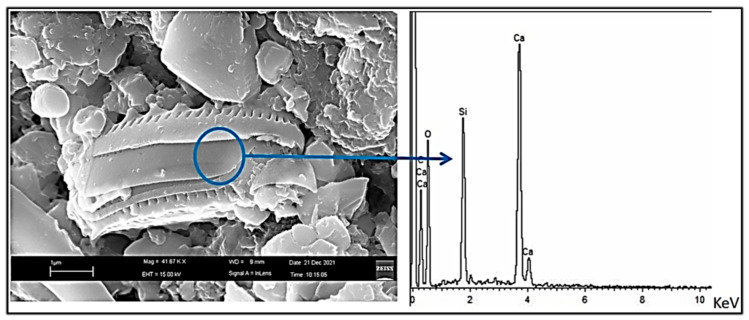
SEM image and associated EDS spectrum of microalgae (diatoms) on the wall.

**Figure 11 molecules-28-00330-f011:**
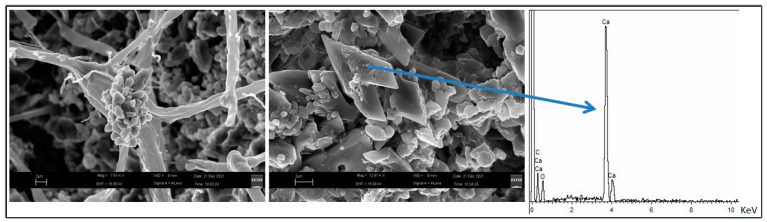
SEM images of calcium carbonate crystals produced by fungal hyphae (**left**) and of calcite (**center**) taken from another area of a fragment free of colored patinas, with-related EDS spectrum (**right**).

**Figure 12 molecules-28-00330-f012:**
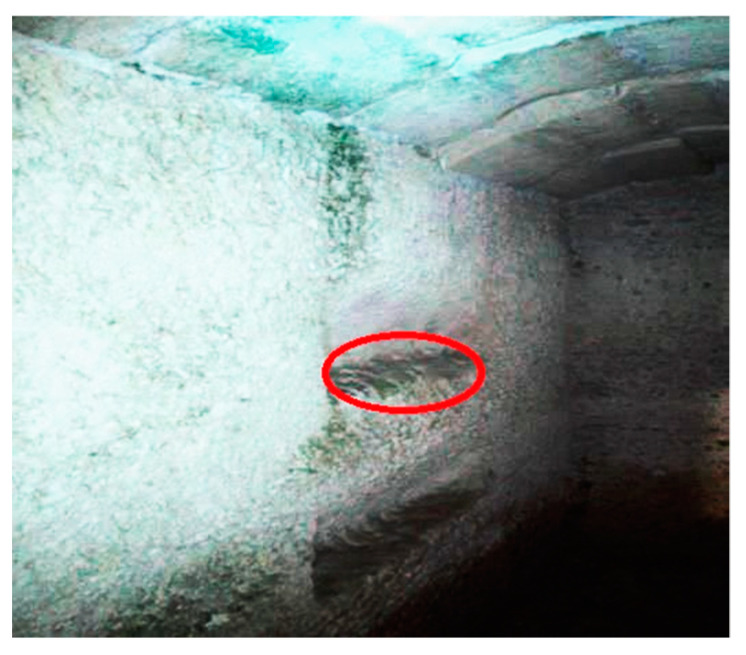
Study area with bio-patina in the hypogeum of San Pietro Barisano church. During the cleaning test, the microclimate conditions of the wall surface portion sampled were locally ranging between 89–92% humidity and 11–16 °C temperature.

**Table 1 molecules-28-00330-t001:** Corrected BE, normalized area, and assignments for C1s regions: pre-treatment sample.

Sample Pre-Treatment				
Region	Peak	B.E. Corrected	Assignments	Normalized Area
C1s	1	283.0	C-C (polycyclics, graphite, carbides)	312.5
	2	283.8	C-C (polycyclics, graphite, carbides)	3157.3
	3	285.2	C-C, C-N	6596.0
	4	286.7	C-OH, φ-OH, C=N	6429.4
	5	288.8	C_2_O_4_^2−^	1877.3
	6	290.0	CO_3_^2−^	5311.7

**Table 2 molecules-28-00330-t002:** Corrected BE, normalized area and assignments for the C1s regions: post-treatment sample.

Sample Post-Treatment			
Peak	B.E. Corrected	Assignments	Normalized Area
1	283.4	C-C (graphite, carbides, polycyclics)	883.0
2	285.0	C-C	4293.0
3	286.1	C-N, C-O, C-O-C	4267.4
4	287.6	C=O, O-C-O	2351.7
5	290.0	CO_3_^2−^	7345.0

**Table 3 molecules-28-00330-t003:** Peak assignment, corrected BE, and normalized area for the Ca2p region.

		Pre-Treatment		Post-Treatment		
Region	Peak	B.E. Corrected	Normalized Area	B.E. Corrected	Normalized Area	Assignments
Ca2p	1	345.4	386.9	345.8	614.8	Ca
	2	347.3	3891.9	347.5	7106.2	CaCO_3_/Ca-alginate
	5	355.4	80.2	355.6	143.3	Shakeup
	6	359.2	80.2	359.3	143.3	Shakeup

**Table 4 molecules-28-00330-t004:** Assignment related to Si2p, P2p, and Na1s peaks.

Region	B.E. Corrected (pre-)	B.E. Corrected (post-)	Normalized Area (pre-)	Normalized Area (post-)	Assignments
Si(2p)	102.7	102.8	608.0	839.5	SiC,SiO_2_
P(2p)	132.7	133.2	58.5	345.1	CaPO_4_
Na(1s)	1072.2	1072.4	387.6	613.7	NaH_2_PO_4_, Na_2_HPO_4_

**Table 5 molecules-28-00330-t005:** Bacterial and fungal species identified in the bio-patina of the hypogeum.

Bacteria	Fungi
*Staphylococcus warneri*	*Botryotrichum atrogriseum*
*Brevibacillus* spp.	*Penicillium chrysogenum*
*Bacillus cereus*	*Talaromyces pinophilus*
*Bacillus mycoides*	*Cladosporium herbarum*
*Bacillus firmus*	

## Data Availability

Not applicable.

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
