# Peer review of "The Bio-Patina on a Hypogeum Wall of the Matera-Sassi Rupestrian Church “San Pietro Barisano” before and after Treatment with Glycoalkaloids"

_molecules, 2022, doi:10.3390/molecules28010330_

Round 1
Reviewer 1 Report
The experimental data need to be refined again, more comments see the attachment.

Author Response
Reviewer 1.
We thank the reviewer for his helpful comments and suggestions. We have tried to respond in the best possible way and make the corrections to the manuscript. We hope we have met all the reviewer's expectations.
Reviewer Comments and Suggestions: The green patina on a hypogeum wall of the Matera-Sassi rupestrian church "San Pietro Barisano" before and after treatment with glycoalkaloids. In this study, sugar alkaloids and solanine extracted from the fruits of St. Peter Barisano were biotreated. The sugar alkaloids were applied to the wall damaged by extensive biological stripes. The research is to select the best glycosyl alkaloid that can control bacteria, fungi, and other causes of wall damage among prevention strategies. This article provides new ideas and methods to solve the problem of wall infringement. The full text has a compact structure, detailed data, and clear regulations. The specific problems are as follows:
Authors comment: to avoid misunderstanding the composition of the green patina, we preferred to modify this definition to green bio-patina or simply bio-patina in the title and throughout the text.
Q 1.2 - How to compare the C1s carbon area before and after treatment with the biocide, the comparative experimental process and data, and what organic components are reduced?
A 1.2 - Despite its complexity, the main components of the bio-patina are shown in the pie charts of Fig. 6. Their relative intensity is derived by the total mass balance of all chemical groups resolved by curve-fitting. As already reported in the text, the carbon 1s regions show the reduction of the peaks at the lower BE side after GAs treatment, which include aliphatic, aromatic, and polycyclic contributions by proteins/peptide-glycans/lipids/ typical of the membrane envelope of bacteria cells [New citation 29 in the text. See reference below] - together with a parallel highlight of the carbonate peak (of the underneath calcarenite matrix). Moreover, the total calcium 2p area is balanced by carbonate and oxalate before treatment. In contrast, after treatment, the oxalate peak disappears, and another component adds to carbonate to balance the calcium area: the carboxylic group of EPS alginates.
[Marit Kjærvik, Madeleine Ramstedt, Karin Schwibbert, Paul M. Dietrich and Wolfgang E.S. Unger. Comparative Study of NAP-XPS and Cryo-XPS for the Investigation of Surface Chemistry of the Bacterial Cell-Envelope. Frontiers in Chemistry (2021) Open Access DOI: 10.3389/fchem.2021.666161]
Q 1.2 - How are bacteria and fungi identified in Figure 9?
A 1.2 - Fungi were identified on the basis of morphological features and confirmed by amplification and sequencing of the two Internal Transcribed Spacers and the 5.8 S gene (ITS1-5.8S-ITS2) from the nuclear ribosomal RNA. The obtained nucleotide sequences were deposited in GenBank (NCBI)and their accession numbers are included in the manuscript.
Bacterial colonies were isolated and characterized based on their different morphologies. The isolated strains were purified, and their DNA was used for molecular identification. Synthetic oligonucleotide primers fD1 (AGAGTTTGATCCTGGCTCAG) and rD1 (AAGGAGGTGATCCAGCC) were used to amplify the 16S rDNA. The PCR products were sequenced using Genetic Analyzer 3130xl (Applied Biosystems), and DNA similarity was performed by the GenBank and EMBL database using the BLAST suite. The related accession numbers are included in the manuscript.
Q 1.3 - How to determine whether calcium oxalate is produced by bacteria or fungi?
A 1.3 - In this regard, the text was incorrectly written that fungi produce oxalate. The correct sentence is now reported in the manuscript “The role of various microorganisms in biofilms is still a complex matter; e.g., it is notable the joint function of compounds such as calcium oxalate, which, although produced by the symbiosis of fungal species and oxalotrophic bacteria, can only be metabolized by the bacteria with the formation of calcium carbonate [see ref. 37]”.
Q 1.4 - Specific microclimate refers to the temperature and humidity.
A 1.4 – The climatic conditions were ascertained and reported in the text “a relative humidity of the surfaces above 90% and an ambient temperature between 11°C and 17°C [9]”. During the cleaning test, the microclimate conditions of the wall surface portion sampled were locally ranging between 89%-92% humidity and 11°C-16°C temperature (see the heading of Fig. 12).
Q 1.5 - What should be paid attention to during the sampling process? What are the requirements for sampling personnel?
A 1.5 - The sampling process should be performed without damaging the site. A non-invasive sampling was performed following the national and International cultural heritage conservation rules. See ref. 42 (cited in the text).
Q 1.6 - What is the temperature of centrifugation during the pretreatment of sugar alkaloid extraction?
A 1.6 - 3000 g for 20 min at 4°C (the temperature value has been added, and “rpm” were recalculated in “g”.
Q 1.7 - To isolate fungi, collect the sample in a sterile vial containing 1ml of double distilled water until analysis. Could you tell me whether the best testing time and too long storage time will be decomposed?
A 1.7 - The samples were stored at 4°C only for a couple of hours and processed within 24 hours from their collection.
Q 1.8 - 3.5 Sequence the amplicon by "Illumina" technology, and then the precautions and operation process in sequencing.
A 1.8 - We corrected the text. The amplicon sequencing was performed using the Sanger Dideoxy technology by an external company, BMR Genomics from Padua, a leader in sequencing in Italy. Therefore, the precautions and operation process in sequencing are well considered by BMR Genomics.
Author Response
Reviewer 2
The manuscript has been revised, but the result does not meet the expectations and basic standards for publication in Molecules which is the leading international, peer-reviewed, open access journal of chemistry having 4,927 IF. Missing linearity, the speech jumps from one item to another without an order, resulting then confusing and inconclusive. In short, the authors should decide which kind of work they want to produce (e.g., ecological, taxonomical, applicative), at present the different parts are disconnected. Moreover, the text evidenced work done in a hurry with the inappropriate use of italics in opening sentences.
Authors answer: Although the reviewer was very strict, we thank him because we have tried to improve our manuscript as much as possible with his stimulus. We hope we have met all the reviewer's expectations.
Below are some additional comments.
Reviewer. The work is intriguing, but its scientific value would be much higher if the authors will focused on biological patina, biodeterioration, cleaning or decontamination.
I recommend to the authors to have in their mind that green patina has different meanings. The green patina that forms naturally on copper and bronze, sometimes called verdigris, usually consists of varying mixtures of copper chlorides, sulfides, sulfates and carbonates, depending upon environmental conditions such as sulfur-containing acid rain. There is also biological patina or a mixture of them. Anyway, patina forms over time on the surface of historical monuments or catacombs. From the manuscript (Fig. 8) it appears that the green biofilm was analysed without identifying the producers. More, diatoms were included. If in green patina had been identified bacteria and fungi (table 5) why it was called green? Again why authors are talking about green patina?
Authors comment: to avoid a misunderstanding about the composition of the green patina, we preferred to modify this definition in ‘green bio-patina’ or simply ‘bio-patina’ in the title and throughout the text.
Reviewer. The authors did not identify the effectiveness of the treatment with glycoalkaloids as a biocide.
Authors comment: To monitor the biocidal action of the glycoalkaloids, samples were taken pre- and post-treatment with the S. Nigrum extract and the modifications in the bio-patina were followed based on the visual appearance and the results of the XPS analysis.
Reviewer. Figure 10 does not have a place in the work because they contain trivial images obtained by SEM
Authors comment: We agree with the reviewer and eliminated this figure.
Reviewer. Subsection 3.3. Bio-cleaning of the hypogeum walls it makes no sense because it is not bioclening but an intention of decontamination.
Authors comment: The subsection 3.3 has been renamed ‘Surface treatment of the hypogeum wall’
Other corrections made:
Row 90 Solanumdulcamara Row 92 Solanumnigrum Rows 221-223 ibidem
Rows 230,231 Penicillium chrysogenumis
Reviewer 3 Report
This is a useful study of biodeterioration of the walls in the hypogeum of “San Pietro Barisano” church. This research work is aimed at finding innovative and environmentally friendly approaches for protection cultural heritage from biodestruction. To assess the effectiveness of the use of a natural biocide based on secondary metabolites Solanum nigrum, the method of XPS photoelectron spectroscopy was used. But the manuscript needs revision. A clearer presentation of one's own research is required. I still recommend separating results from discussion.
The topic presented by the authors in the manuscript seems interesting and useful. I would like to see the manuscript published, but after a major revision.
I have the following comments which I hope will be helpful to the authors
Some paragraphs and sentences may need to be moved to the materials and methods section. Such as: Lines 236-241 – laboratory test of biocidal activity of glycoalkaloids of Solanum nigrum on fungi B. atrogriseum and T. cespinophilus.
In section 3.5 The authors described in details the methods of isolation and identification of bacteria and fungi, but pay no attention to algae. The green biofilm on the walls of the hypogeum is formed by algae. For fungi in this section 3.5, the nutrient medium (PDA) on which they were isolated is given. But such a medium is not given for bacteria.
In section 2.2.The green biofilm on the walls of the hypogeum is formed by algae. What about algae, what taxonomic group do they belong to. Maybe among them there are cyanobacteria? If there is information about the taxonomic groups of algae and cyanobacteria, it should also be given. Only in section 2.3, the authors indicate the presence of diatom cells on the SEM image.
Figure 7. Graphic images are too small. The numbers are very hard to see.
Figure 8. The photos in Figure 8 are out of focus. Is it possible to put better photos?
Figure 11, Figure 12. Images are too small, EDS spectrum - symbols of chemical elements are not visible on the spectrum.
Figure 10 Image details of fungal hyphae and the distribution of microorganisms on the surface of the sample are not very visible. Is it possible to put better photos?
Figure 12. On the EDS spectrum, the elements calcium, carbon and oxygen can hardly be seen. But this does not prove that the crystals are calcite and not calcium oxalate. In their composition (calcite and calcium oxalate) are the same elements. X-ray phase analysis data could more accurately answer the question about the nature of these crystals. Fungi of the genus Penicillium can produce both calcium oxalates and carbonates. And the authors isolated and identified Penicillium chrysogenum, Talaromyces pinophilus from samples of green patina on wall.
Figure 13. The photo is out of focus. Is it possible to put better photo?
Lines 242-243. The authors described write that fungi can used dead bacterial cells as a carbon source. What about algae dead cells? In this green biofilm on the walls of the hypogeum algae and maybe cyanobacteria dominate.
Lines 302-303. What other components?
And small remarks on the text:
Lines 90, 92, 210, 222, 223, 231,
Line 237. Solanum nigrum L. have the author species, but the all fungi (as example Penicillium chrysogenum…) have not the authors species. It is necessary either to remove the author of the species from Solanum nigrum, or to put the authors of the species to all fungi.
Lines 498-500 Remove space
Author Response
Reviewer 3
We thank the reviewer for his helpful comments and suggestions. We have tried to respond in the best possible way and make the corrections to the manuscript. We hope we have met all the reviewer's expectations.
Reviewer Comments and Suggestions: This is a useful study of biodeterioration of the walls in the hypogeum of “San Pietro Barisano” church. This research work is aimed at finding innovative and environmentally friendly approaches for protection cultural heritage from biodestruction.To assess the effectiveness of the use of a natural biocide based on secondary metabolites Solanum nigrum, the method of XPS photoelectron spectroscopy was used. But the manuscript needs revision. A clearer presentation of one's own research is required. I still recommend separating results from discussion.
The topic presented by the authors in the manuscript seems interesting and useful. I would like to see the manuscript published, but after a major revision.
I have the following comments which I hope will be helpful to the authors
Authors comment: to avoid misunderstanding the composition of the green patina, we preferred to modify this definition to green bio-patina or simply bio-patina in the title and throughout the text.
Q 1 - Some paragraphs and sentences may need to be moved to the materials and methods section. Such as: Lines 236-241 – laboratory test of biocidal activity of glycoalkaloids of Solanum nigrum on fungi B. atrogriseum and T. pinophilus.
A 1 – Thank you for this suggestion, but we cannot move this paragraph that contains mainly results obtained with a laboratory test.
Moreover, the results are organized in subsections that need a prompt discussion to keep the attention of the reader who can see the data and read the related interpretation.
Q 2.1 - In section 3.5 The authors describedin details the methods of isolation and identification of bacteria and fungi, but pay no attention to algae. The green biofilmon the walls of the hypogeum is formed by algae.
A 2.1 – In a previous article cited in the manuscript as ref. 17 we found Bryophytes and lichens, primary colonizers producing organic material and serving as a substrate for secondary colonists, on the internal walls of the church. In particular, Grimmia pulvinata and Tortula muralis. It was impossible to clearly identify algal species due to poor morphological characteristics detectable.
Q 2.2 - For fungi in this section3.5, the nutrient medium(PDA) on which they were isolated is given. But such a medium is not given for bacteria.
A 2.2 – We have added the needed information in the revised manuscript
Q 3 - In section 2.2. The green biofilmon the walls of the hypogeum is formed by algae. What about algae, what taxonomic group do they belong to. Maybe among them there are cyanobacteria? If there is information about the taxonomic groups of algae and cyanobacteria, it should also be given. Only in section 2.3, the authors indicate the presence of diatom cells on the SEM image.
A 3 – Sorry, we could not clearly identify algal species or ascertain the presence of cyanobacteria in the tested area. Surely, these organisms are colonizing other parts of the church walls characterized by more lighting. The site tested was in a darker sector of the church. We selected this sector to avoid the overcrowding of different organisms — our experiment aimed to reduce bacterial and fungal species utilizing a natural product. Moreover, we wanted to study the eventual correspondence of chemical and instrumental data with the biological evidence before and after the bio-treatment.
Figure 7. Graphic images are too small. The numbers are very hard to see. Done
Figure 8. The photos in Figure 8 are out of focus. Is it possible to put better photos? We tried to improve the photo
Figure 11, Figure 12. Images are too small, EDS spectrum - symbols of chemical elements are not visible on the spectrum. Done (this figure has been renamed as Fig. 11)
Figure 10 Image details of fungal hyphae and the distribution of microorganisms on the surface of the sample are not very visible. Is it possible to put better photos? This figure has been eliminated.
Q 4 - Figure 12. On the EDSspectrum, the elements calcium, carbon and oxygen can hardly be seen. But this does not prove that the crystals are calcite and not calcium oxalate. In their composition (calcite and calcium oxalate) are the same elements. X-ray phase analysis data could more accurately answer the question about the nature of these crystals. Fungi of the genus Penicillium can produce both calcium oxalates and carbonates. And the authors isolated and identified Penicillium chrysogenum, Talaromyces pinophilus from samples of green patina on wall.
A 4 - With the repeated bombardment of the electron beam (energy source) on the sample, the oxalate crystals can dissolve and change their shape, which did not happen in this case. In fact, the images in Fig. 12 (renamed as Fig. 11) have remained unchanged over time, a typical situation of calcium carbonate crystals.
Figure 13. The photo is out of focus. Is it possible to put better photo? We tried to improve the photo (Renamed as Fig. 12). It was a dark site.
Q 5 - Lines 242-243. The authors described write that fungi can use dead bacterial cells as a carbon source. What about algae dead cells? In this green biofilm on the walls of the hypogeum algae and maybe cyanobacteria dominate.
A 5 – The experimented site (Fig. 12) was in darkness. It does not present evident colonies of autotrophy organisms.
Q 6 - Lines 302-303. What other components?
A 6 – we added the citation of two previous articles we published (References 23 and 36). A more accurate determination of minor components is reported in a new article to be submitted next month.
And small remarks on the text:
Lines 90, 92, 210, 222, 223, 231, revised
Q 7 - Line 237. Solanum nigrum L. have the author species, but the all fungi (as example Penicilliumchrysogenum…) have not the authors species. It is necessary either to remove the author of the species from Solanum nigrum, or to put the authors of the species to all fungi.
A 7 - As suggested by the reviewer, the author was removed from the S. nigrum species.
Lines 498-500 Remove space
As suggested, the space was removed.
Round 2
Reviewer 3 Report
I appreciate that the authors have addressed all of my comments and I find this version improved from the previous one. I have a comment in response to the author's responses.
Replacement the term green patina with the term bio-patina in the text of the manuscript seems to me more successful options. It may be better to remove the word “green” from the title of the manuscript and leave only a bio-patina. Thank you for including more details and figures in better quality.
The section «Ð¡onclusions». Please reduce this section. Don’t include discussions here. Highlight only important conclusions.
Author Response
Thank you very much for your final evaluation and suggestions.
We replaced the words 'green bio-patina' and 'green patina' with 'bio-patina' in the title and throughout the text. Moreover, we reduced the 'Conclusions' section and eliminated the repetition of citations.